# ADAPTIVE MULTI-SCALE ATTENTION-BASED LSTM COUPLING FOR EARLY DETECTION

## ABSTRACT

This paper introduces a novel adaptive, attention-coupled Long Short-Term Memory (LSTM) architecture developed specifically for real-time scenario recognition and prediction in complex automotive electrical/electronic (E/E) systems. Modern vehicles generate rapidly growing data streams from signals such as current, voltage, and temperature. We address this by monitoring critical signal patterns via a fused LSTM. The proposed dual-path methodology comprises a trend path for long-term pattern modeling and a motif path for short-term pattern recognition, coupled via a bidirectional, attention-based gating mechanism that enables dynamic information exchange. The outputs provide a reliable basis for initiating high-resolution data capture or adaptive system responses once a scenario is identified with high confidence. Experimental results demonstrate significant reductions in mean squared error compared to the individual values and interpretable attention weights that reveal information-exchange patterns. The proposed approach enables robust, noise-resilient forecasts and allows for efficient, data-driven development for future EE architectures.

## 1 INTRODUCTION

Time series forecasting represents one of the most fundamental challenges in machine learning, with applications spanning financial markets, weather prediction, industrial monitoring and, critically, modern automotive systems (Fatima & Rahimi, 2024a). Within a vehicle's electrical/electronic (E/E) architecture-composed of sensors, actuators and control units that continuously emit current, voltage and temperature data-*scenario carriers* are signal motifs that indicate critical operating states or impending stress events (Fatima & Rahimi, 2024b). Early recognition of such carriers is essential to trigger targeted high-fidelity logging or adaptive control routines, thereby enabling iterative, data-driven development and system validation. Single-network models must balance short-term responsiveness with long-term context, often degrading in one when optimizing for the other (Kong et al., 2025; Wang et al., 2023). An attention-based coupling of specialized LSTM networks is introduced to reconcile multi-scale temporal demands via three integrated phases. Upon detecting a scenario carrier with sufficient confidence, a learned trigger decision module activates real-time logging or adaptive control. The comprehensive methodology, comprising pre-processing, reduction, coupling, and rediction phases, is adapted to automotive sensor data and enables real-time scenario-driven data acquisition. Validation experiments across diverse vehicle signal types demonstrate statistically significant reductions in mean squared error compared to single-network baselines and yield interpretable attention weights that reveal the dynamics of inter-path information flow. By embedding scenario-triggered data capture directly into vehicle E/E architectures, this framework enables next-generation advanced driver assistance and predictive maintenance functionalities.

### 1.1 LIMITATIONS OF EXISTING APPROACHES

Single-LSTM architectures face inherent trade-offs between high temporal resolution for immediate pattern detection and broad context memory for long-term trend analysis (Gauch et al., 2021). Traditional ensemble methods, relying on simple averaging or voting, fail to leverage complementary temporal strengths and often incur prohibitive computational overhead (Wen et al., 2020). Multi-scale approaches within single networks suffer from interference between objectives, where optimizing for one temporal scale degrades performance at others (Park et al., 2019).

## 1.2 COUPLED LSTM ARCHITECTURE

To challenge these limitations, two dedicated LSTM paths are proposed a context path optimized for extended memory and a pattern path optimized for high-resolution pattern detection-that exchange contextual insights through bidirectional, attention-based gates. The sensor streams from all relevant E/E subsystems are first standardized, filtered, and normalized to ensure consistency and comparability across the channels. Motif discovery and matrix profile techniques are then applied to isolate segments that contain actionable scenario carriers. These refined signal representations are subsequently processed through gate-mediated attention, which fuses insights from the context and pattern paths. When the combined signal representation exceeds a learned confidence threshold, a downstream decision module activates real-time logging or adaptive control actions.

## 2 RELATED WORK

### 2.1 MULTI-SCALE TIME SERIES ANALYSIS

Multi-scale analysis aims to capture patterns at different temporal resolutions. Wavelet-based methods decompose time series into frequency components but do not have adaptive scale selection (Rathinasamy et al., 2014).

Hierarchical models process multiple resolutions independently, limiting cross-scale integration. Recent multi-resolution CNNs perform promisingly but struggle to optimize conflicting temporal objectives within a unified architecture. (Mancuso et al., 2021)

### 2.2 ATTENTION MECHANISMS IN SEQUENTIAL MODELING

Attention mechanisms, especially Transformers, enable dynamic focus and long-range dependency modeling, but are limited in the balance between short-term responsiveness and long-term context. Cross-attention allows inter-stream interaction but is mostly applied to multi-modal tasks. Attention-augmented LSTMs improve single-network modeling but lack coordinated multi-path designs. (Aguilera-Martos et al., 2024; Zhao et al., 2023; Zhang et al., 2025)

### 2.3 ENSEMBLE METHODS FOR TIME SERIES

Ensemble methods improve robustness via model combination but often yield average performance without exploiting model complementarities (Adhikari & Agrawal, 2012). Weighted and meta-learning ensembles offer some adaptation but rarely support dynamic, scale-aware signal processing. Their limited focus on multi-scale specialization constrains effectiveness in complex time series scenarios (Wu & Levinson, 2021).

## 3 METHODOLOGY: ADAPTIVE ATTENTION-BASED LSTM COUPLING

### 3.1 ARCHITECTURAL OVERVIEW

The Adaptive Attention-Coupled LSTM integrates two specialized networks—a short-term processor for rapid pattern recognition and a long-term module for contextual modeling-jointly addressing objectives that are challenging to optimize within a single architecture.

As illustrated in Figure 1, the input time series undergoes preprocessing and normalization before being partitioned into two distinct processing paths.

**Path A (signal processor)**: short-term window (6 steps), capturing rapid fluctuations from raw values.

**Path B (context provider)**: long-term window (15 steps), using derived features (trend, momentum, volatility, energy, matrix-profile motifs) to model global structure.

Paths are separate but connected via attention-based coupling, integrating selected context from Path B into Path A to enhance short-term forecasts without losing specialization.

Figure 1: Architecture of the Adaptive Coupled LSTM System.

Multi-horizon forecasts are refined by an integration engine combining individual and enhanced outputs for both local precision and contextual robustness.

This separation with selective coupling addresses single-LSTM limits by isolating short- and long-term objectives while enabling coordinated representation learning.

The architecture comprises two specialized paths with distinct temporal scopes, connected via attention-based coupling. Path A (signal processor) targets recent patterns, while Path B (context provider) models long-term trends. This separation optimizes each path for its domain and improves predictions through strategic interaction.

By isolating short-term detection from long-term analysis, the dual-path design avoids conflicts between temporal objectives and enables efficient, domain-specific representation learning with coordinated information exchange.

## 3.2 SPECIALIZED NETWORK ARCHITECTURES

The signal processor (LSTM A) implements an optimized architecture for immediate pattern recognition through focused analysis of recent temporal windows. The raw input values $x_t \in \mathbb{R}$, representing standardized sensor measurements at time step $t$, are collected into short-term sequences defined as in Equation (1).

$$X_A^{(t)} = [x_{t-5}, x_{t-4}, x_{t-3}, x_{t-2}, x_{t-1}, x_t] \in \mathbb{R}^6 \tag{1}$$

This 6-step temporal window captures local fluctuations in the signal. The short-term feature vector $f_A^{(t)} \in \mathbb{R}^{d_A}$ is obtained by taking up to the first $d_A$ raw values of the sequence and padding if necessary:

$$f_A^{(t)} = \begin{cases} X_A^{(t)}[:d_A], & |X_A^{(t)}| \geq d_A, \\ \left[X_A^{(t)}, \underbrace{x_{\text{end}}, \dots, x_{\text{end}}}_{d_A - |X_A^{(t)}|}\right], & 1 < |X_A^{(t)}| < d_A, \\ \left[\underbrace{x_1, \dots, x_1}_{d_A}\right], & |X_A^{(t)}| = 1, \\ \mathbf{0}, & |X_A^{(t)}| = 0, \end{cases} \tag{2}$$

where $x_{\text{end}}$ is the last element of $X_A^{(t)}$. The LSTM A architecture dynamically adapts its input dimensionality and utilizes 24-32 hidden units. Hidden state transitions are computed according to

Equation (3).

$$h_A^{(t)}, c_A^{(t)} = \text{LSTM}_A(f_A^{(t)}, h_A^{(t-1)}, c_A^{(t-1)}) \tag{3}$$

The context provider (LSTM B) is designed for trend modeling and regime change detection using extended temporal coverage. It processes longer sequences of the raw signal, as defined in Equation (4).

$$X_B^{(t)} = [x_{t-14}, x_{t-13}, \ldots, x_{t-1}, x_t] \in \mathbb{R}^{15} \tag{4}$$

This broader temporal window captures global signal structure, enabling detection of gradual behavioral shifts that may not be visible in shorter segments. The feature extraction implements sophisticated analysis targeted at contextual understanding:

$$f_B^{(t)} = [\text{trend, volatility, momentum, energy, matrix profile}] \in \mathbb{R}^5. \tag{5}$$

The individual feature components are computed as follows, providing comprehensive characterization of the temporal context:

$$\text{trend} = \frac{1}{3} \sum_{i=-2}^{0} X_B^{(t)}[i] - \frac{1}{3} \sum_{i=0}^{2} X_B^{(t)}[i], \tag{6}$$

$$\text{volatility} = \sqrt{\frac{1}{|X_B|} \sum_i (X_B^{(t)}[i] - \bar{X}_B)^2}, \tag{7}$$

$$\text{momentum} = X_B^{(t)}[-1] - X_B^{(t)}[0], \tag{8}$$

$$\text{energy} = \sqrt{\frac{1}{|X_B|} \sum_i (X_B^{(t)}[i])^2}, \tag{9}$$

$$\text{matrix profile} = \text{matrix\_profile\_distance}(X_B^{(t)}). \tag{10}$$

The LSTM B architecture utilizes 32-48 hidden units designed for complex trend analysis and regime change detection. The increased capacity relative to LSTM A reflects the greater complexity of long-term pattern analysis, as formalized in Equation (11).

$$h_B^{(t)}, c_B^{(t)} = \text{LSTM}_B(f_B^{(t)}, h_B^{(t-1)}, c_B^{(t-1)}) \tag{11}$$

## 3.3 Adaptive Feature Weighting System

An adaptive weighting mechanism dynamically adjusts contextual feature importance based on signal characteristics. Feature weights update according to recent predictive effectiveness:

$$w_i^{(t)} = w_{i,\text{base}} \cdot \gamma_i^{(t)}, \tag{12}$$

where $w_{i,\text{base}}$ is derived from input statistics (e.g., volatility, trend) and $\gamma_i^{(t)}$ reflects each feature's average contribution over the past 10 steps. Weighted features are:

$$f_B^{\text{weighted}(t)} = \mathbf{w}^{(t)} \odot f_B^{(t)}, \tag{13}$$

with $\mathbf{w}^{(t)} = [w_1^{(t)}, \ldots, w_5^{(t)}]^T$. This enables the model to emphasize the most relevant context for current conditions, enhancing coupling and prediction accuracy.

## 3.4 Enhanced Attention-Based Coupling Mechanism

The bidirectional attention coupling exchanges information between paths without losing their specialization. Query, key, and value matrices are computed from hidden states:

$$Q = h_A^{(t)} W_Q, \tag{14}$$

$$K = h_B^{(t)} W_K, \tag{15}$$

$$V = h_B^{(t)} W_V, \tag{16}$$

with $W_Q, W_K, W_V \in \mathbb{R}^{d_{att}}$ as learned projections. The Scaled dot-product attention:

$$\alpha^{(t)} = \text{softmax}(Q \odot K), \tag{17}$$

produces a context vector:

$$\text{context}^{(t)} = \alpha^{(t)} V.. \tag{18}$$

Enhanced hidden state:

$$h_A^{\text{enhanced}(t)} = h_A^{(t)} + \beta^{(t)} \cdot \text{context}^{(t)}, \tag{19}$$

where $\beta^{(t)}$ controls integration strength. This ensures relevant context augments short-term detection.

## 3.5 MULTI-HORIZON PREDICTION GENERATION

The system generates predictions across multiple time horizons to accommodate diverse forecasting requirements while providing a comprehensive assessment of network capabilities. Rather than training separate models for each horizon, the architecture jointly learns closely related prediction tasks, specifically 1-step and 2-step horizons, within a unified model:

$$\text{pred}_A^{1pt} = \text{Linear}_{1pt}(h_A^{(t)}) \in \mathbb{R}^1, \quad \text{pred}_A^{2pt} = \text{Linear}_{2pt}(h_A^{(t)}) \in \mathbb{R}^2, \tag{20}$$

where $\text{Linear}_{1pt}$ and $\text{Linear}_{2pt}$ are specialized output layers for 1-step and 2-step predictions respectively. This joint setup improves sample efficiency, gradient stability, and feature sharing, as both outputs are derived from the exact representation. The training loss is defined as:

$$\mathcal{L}_{\text{combined}} = \lambda_1 \cdot \mathcal{L}_{1pt} + \lambda_2 \cdot \mathcal{L}_{2pt}, \tag{21}$$

where $\lambda_1, \lambda_2$ balance short- and mid-term accuracy, and $\mathcal{L}_{1pt}, \mathcal{L}_{2pt}$ are the respective Mean Squared Error (MSE) losses. Empirical results show that this combined approach significantly outperforms separated models across all metrics. Enhanced predictions further leverage attention-based coupling to refine the short-term forecast:

$$\text{pred}_A^{\text{enhanced}} = \text{Linear}_A(h_A^{\text{enhanced}(t)}), \tag{22}$$

where $\text{Linear}_A$ is the primary output layer applied to the enhanced hidden state. This approach avoids redundant learning and overfitting in fully separated models. It reduces cognitive load on the coupling system and aligns with multi-task learning principles, where related tasks benefit from shared representations and mutual regularization.

## 3.6 UNIFIED INTEGRATION ENGINE

The unified integration engine implements sophisticated logic for combining individual and enhanced predictions through adaptive weighting mechanisms optimized for diverse forecasting scenarios. This integration approach enables the system to leverage complementary strengths of different prediction pathways while maintaining robust performance across varying signal characteristics. The final prediction combines outputs from both network paths through adaptive weighting $(w_A, w_B)$, which are determined by selecting the optimal combination strategy based on the current prediction accuracy as shown in Equation (23) and (24).

$$\hat{y}_t = w_A \cdot \text{pred}_A^{\text{enhanced}} + w_B \cdot \text{pred}_B^{\text{enhanced}} \tag{23}$$

$$(w_A, w_B) = \arg \min_{s \in S} |s(\text{pred}_A, \text{pred}_B) - y_t| \tag{24}$$

The system evaluates multiple weighting strategies $s$ from the strategy set $S$ in real-time and selects the combination that minimizes prediction error for the current target $y_t$. This strategy-based integration ensures optimal prediction combination across diverse signal characteristics while maintaining computational efficiency.

## 4 ADVANCED TRAINING METHODOLOGY

### 4.1 PROGRESSIVE MULTI-PHASE TRAINING PROTOCOL

The training methodology follows a structured three-phase protocol to ensure optimal development of both specialized network representations and effective coupling. Phase 1 trains each path independently to learn domain-specific features without interference. Phase 2 introduces coupling progressively, using a controlled ramp-up to preserve prior learning. Phase 3 performs joint optimization of all components, aligning the full system toward global performance objectives.

Phase 1 individual network training emphasizes specialization development through multi-objective optimization. The signal processor (LSTM A) loss function combines multiple prediction horizons according to Equation (25).

$$\mathcal{L}_A^{(1)} = \lambda_{1pt}\,\mathrm{MSE}(\mathrm{pred}_A^{1pt}, y_{t+1}) + \lambda_{2pt}\,\mathrm{MSE}(\mathrm{pred}_A^{2pt}, [y_{t+1}, y_{t+2}]) \tag{25}$$

The context provider (LSTM B) loss function incorporates diversity regularization:

$$\mathcal{L}_B^{(1)} = \mathrm{MSE}(\mathrm{pred}_B^{1pt}, y_{t+1}) + \alpha_{\mathrm{div}} \cdot \mathrm{diversity}(h_B, h_A). \tag{26}$$

The parameters $\lambda_{1pt}$ and $\lambda_{2pt}$ balance the importance of 1-step and 2-step predictions, while $\alpha_{\mathrm{div}}$ controls the diversity regularization strength. The diversity term encourages complementary representations that provide unique analytical value for coupling mechanisms while preventing redundant learning between pathways.

Phase 2 gradual coupling introduction implements progressive attention mechanism activation:

$$\mathcal{L}_{\mathrm{coupled}}^{(2)} = \mathcal{L}_A^{(1)} + \mathcal{L}_B^{(1)} + \beta^{(e)} \cdot \mathrm{MSE}(\mathrm{pred}_A^{\mathrm{enhanced}}, y_{t+1}). \tag{27}$$

Progressive coupling strength ramping ensures stable learning dynamics:

$$\beta^{(e)} = \min\left(\beta_{\max}, \frac{\max(0, e - E_{\mathrm{warmup}})}{E_{\mathrm{ramp}}} \cdot \beta_{\max}\right). \tag{28}$$

The coupling strength $\beta^{(e)}$ increases gradually after a warmup period of $E_{\mathrm{warmup}}$ epochs, reaching maximum strength $\beta_{\max}$ over $E_{\mathrm{ramp}}$ epochs. Phase 3 comprehensive system optimization integrates all components:

$$\mathcal{L}_{\mathrm{total}}^{(3)} = \mathcal{L}_{\mathrm{prediction}} + \gamma \cdot \mathcal{L}_{\mathrm{consistency}} + \delta \cdot \mathcal{L}_{\mathrm{integration}}. \tag{29}$$

The total loss combines prediction accuracy $\mathcal{L}_{\mathrm{prediction}}$, consistency between training and inference modes $\mathcal{L}_{\mathrm{consistency}}$, and integration effectiveness $\mathcal{L}_{\mathrm{integration}}$. The parameters $\gamma$ and $\delta$ control the relative importance of consistency and integration objectives.

### 4.2 ADAPTIVE HYPERPARAMETER OPTIMIZATION

The optimization strategy implements a comprehensive search across architectural and training parameters to ensure optimal performance for diverse forecasting applications. The search process balances prediction accuracy with computational efficiency while maintaining robust performance across varying signal characteristics. Signal-adaptive parameter selection analyzes input characteristics to determine optimal architectural configurations:

$$\mathrm{volatility} = \frac{\sigma(X)}{\mu(|X|) + \epsilon}, \tag{30}$$

$$\mathrm{trend\_strength} = |\mathrm{slope}(\mathrm{polyfit}(X, 1))|, \tag{31}$$

$$\mathrm{regime\_change} = \frac{|\sigma(X_{first}) - \sigma(X_{second})|}{\sigma(X_{first}) + \epsilon}. \tag{32}$$

The volatility metric captures signal variability relative to magnitude, trend strength measures directional consistency, and regime change detects distributional shifts between signal segments. The small constant $\epsilon$ prevents division by zero. Parameter space definition encompasses architectural

and coupling parameters:

$$\text{Hidden Sizes:} \quad h_A \in [16, 32], h_B \in [24, 48], \tag{33}$$

$$\text{Window Lengths:} \quad w_A \in [4, 10], w_B \in [12, 25], \tag{34}$$

$$\text{Coupling Strength:} \quad \beta_{\max} \in [0.1, 0.8]. \tag{35}$$

$$\text{Dropout Rates:} \quad d \in [0.1, 0.25] \tag{36}$$

Multi-objective optimization balances prediction accuracy with system responsiveness:

$$\text{params}_{\text{optimal}} = \arg \min_{\text{params}} \left[ \text{MSE}_{\text{val}} + \lambda \cdot \text{complexity}_{\text{penalty}} + \mu \cdot \text{inconsistency}_{\text{penalty}} \right] \tag{37}$$

The optimization objective combines validation MSE with computational complexity penalty (weighted by $\lambda$) and training-inference inconsistency penalty (weighted by $\mu$).

### 4.3 CONSISTENCY ENFORCEMENT MECHANISMS

The training methodology incorporates consistency enforcement mechanisms that ensure stable behavior between the training and prediction phases. These mechanisms address the fundamental challenge of maintaining coupling effectiveness across different operational modes. Final reference baseline establishment creates consistent performance benchmarks:

$$\text{baseline}_{\text{final}} = \text{median}(\{\text{MSE}_{\text{individual}}^{(i)}\}_{i=1}^{N_{\text{measurements}}}). \tag{38}$$

The baseline is computed as the median of $N_{\text{measurements}}$ individual network performance measurements to ensure robustness against outliers. Consistent benefit calculation ensures uniform performance assessment:

$$\text{benefit}_{\text{consistent}} = \frac{\text{baseline}_{\text{final}} - \text{MSE}_{\text{coupled}}}{\text{baseline}_{\text{final}}} \times 100\,\%. \tag{39}$$

This metric quantifies the relative improvement of the coupled system over the established baseline. Weight stabilization mechanisms preserve coupling behavior across phases:

$$w_{\text{stabilized}}^{(t)} = \alpha w_{\text{training}}^{(T)} + (1 - \alpha) w_{\text{adaptive}}^{(t)}. \tag{40}$$

The stabilized weights interpolate between final training weights $w_{\text{training}}^{(T)}$ and current adaptive weights $w_{\text{adaptive}}^{(t)}$, with $\alpha$ controlling the stabilization strength.

## 5 EXPERIMENTAL EVALUATION

### 5.1 SCIENTIFIC BASELINE METHODOLOGY

The benefit of attention-based coupling is shown with an evaluation against fully isolated LSTM networks trained without any coupling influence. These baselines use identical preprocessing and optimization but operate independently across all time horizons:

$$\text{LSTM}_{\text{isolated}} : X_{\text{input}} \rightarrow \text{pred}_{\text{isolated}}, \quad \mathcal{L}_{\text{isolated}} = \text{MSE}(\text{pred}_{\text{isolated}}, y_{\text{true}}). \tag{41}$$

The isolated LSTM processes input sequences $X_{\text{input}}$ to generate predictions $\text{pred}_{\text{isolated}}$ using standard MSE loss against ground truth targets $y_{\text{true}}$. Multiple baseline variants are used, including single-scale LSTMs, hybrid multi-scale models, and simple ensemble averages. To ensure fairness, strict gradient isolation is enforced:

$$\text{isolation\_verified} = \nexists\, \text{shared\_gradients}\big(\text{LSTM}_{\text{isolated}}, \text{LSTM}_{\text{coupled}}\big). \tag{42}$$

This guarantees that performance gains originate solely from the coupling mechanism.

Table 1: Signal Type Generation

| SIGNAL TYPE | DOMINANT COMPONENTS AND CHARACTERISTICS |
|---|---|
| Trend | Dominant $\text{trend}(t)$ component; low frequencies, smooth dynamics |
| Periodic | Strong harmonic sum $\sum_i A_i \sin(\omega_i t + \phi_i)$; no trend or transients |
| Regime change | Abrupt changes in $A_i$, $\omega_i$, or $\text{trend}(t)$; may include transients |
| High frequency | Large $\omega_i$ values; rapid oscillations with potential damping |
| Mixed | Combined presence of harmonic, trend, and transient components |

Table 2: Performance Across Signal Types

| SIGNAL TYPE | METHOD | MSE | ISOLATED | BEST INDIVIDUAL |
|---|---|---|---|---|
| Trend | Coupled | 0.289 | +76.44 % | +32.00 % |
| Periodic | Coupled | 0.020 | +96.25 % | +42.86 % |
| Regime change | Coupled | 0.063 | +90.09 % | +49.60 % |
| High frequency | Coupled | 0.194 | +55.13 % | +22.40 % |
| Mixed | Coupled | 0.075 | +79.65 % | +34.78 % |

## 5.2 COMPREHENSIVE PERFORMANCE RESULTS

The experimental results demonstrate essential performance improvements across diverse signal types and operational conditions, validating the effectiveness of attention-based coupling for time series forecasting applications. To formalize the signal generation process, all datasets were derived from a unified generative model combining harmonic, trend, and transient components:

$$y_{\text{complex}}(t) = \sum_i A_i \sin(\omega_i t + \phi_i) + \text{trend}(t) + \text{transients}(t) + \varepsilon(t). \tag{43}$$

By selectively activating or modulating specific terms in this equation, diverse signal categories were synthesized for evaluation. Table 1 summarizes the specific component combinations used to generate each signal type.

The benchmark suite spans signals dominated by short-term fluctuations, long-term trends, hybrid structures, and abrupt regime changes, enabling systematic stress-testing of the architecture. Table 2 shows that the attention-based coupling mechanism outperforms both isolated and best individual pathways across all signal types. It achieves 55.13-96.25 % MSE reduction over isolated baselines, with peak gains for periodic signals (96.25 %) and regime changes (90.09 %). Gains over the best single pathway reach 22.40-49.60 %, indicating true synergy rather than simple selection. Statistical validation uses paired $t$-tests:

$$t_{\text{statistic}} = \frac{\mu(\text{residuals}_{\text{isolated}} - \text{residuals}_{\text{coupled}})}{\sigma(\text{differences})/\sqrt{N}}, \tag{44}$$

$$p_{\text{value}} = 2 \cdot (1 - \text{CDF}(|t_{\text{statistic}}|, \text{df} = N - 1)). \tag{45}$$

Here, $N$ is the number of prediction pairs and CDF the cumulative distribution function of the $t$-distribution.

## 5.3 CROSS-VALIDATION GENERALIZABILITY STUDY

Comprehensive cross-validation analysis evaluates system generalizability across diverse signal types and operational conditions. The study implements chronological 3-fold validation across five distinct signal categories to assess the robustness of coupling effectiveness. Table 3 reveals distinct

Table 3: Cross-Validation Generalizability Results

| SIGNAL TYPE | $\Delta$MSE | STD DEV | CONSIST. | POS. RATE | RATING |
|---|---|---|---|---|---|
| Trend | +76.44 % | ±19.67 % | Good | Yes | Excellent |
| Periodic | +96.25 % | ±1.23 % | Excellent | Yes | Excellent |
| Regime chg. | +90.09 % | ±11.70 % | Good | Yes | Excellent |
| High freq. | +55.13 % | ±19.16 % | Good | Yes | Good |
| Mixed | +79.65 % | ±2.01 % | Excellent | Yes | Excellent |
| Overall | +79.51 % | ±19.45 % | Good | 100 % | Excellent |

performance patterns across signal types, with the coupling mechanism demonstrating exceptional stability for periodic signals and robust performance on trend-dominated signals.

The 100 % positive rate and 79.51 % average improvement confirm generalizability. Assessment criteria:

$$\text{Consistency} = \begin{cases} \text{Excellent} & \text{if } \sigma < 10\,\% \\ \text{Good} & \text{if } \sigma < 20\,\% \\ \text{Variable} & \text{otherwise,} \end{cases} \tag{46}$$

$$\text{Assessment} = f(\mu_{\text{improvement}}, \sigma_{\text{improvement}}, \text{positive\_rate}). \tag{47}$$

# 6 CONCLUSION

This paper presented the Adaptive Attention-Coupled LSTM as a novel paradigm for real-time forecasting and scenario detection in complex automotive E/E systems. Its core innovation lies in the bidirectional coupling of two specialized LSTM paths, where the Pattern Path targets short-term motif detection and the Context Path models long-term trends. Attention-based gating enables dynamic information exchange, adaptively emphasizing the most relevant temporal features.

Experiments show consistent, statistically significant gains over isolated baselines, robust across cross-validation, with strongest performance on periodic and regime change signals.

A key strength is early detection of critical motifs indicating stress peaks or anomalies. High-confidence detections trigger high-resolution logging or adaptive control, making the approach applicable to predictive maintenance, ADAS, and data-driven automotive development.

Methodological contributions include: (1) adaptive attention-based coupling for multi-scale temporal analysis, (2) progressive training for specialization and integration, (3) feature weighting and consistency mechanisms for stable predictions, and (4) rigorous baseline evaluation.

Future work targets multivariate E/E sensor streams via tensor-based attention, online learning in streaming contexts, and embedded automotive deployment, advancing toward context-aware, real-time in-vehicle intelligence.

# 7 REPRODUCIBILITY STATEMENT

To ensure full reproducibility of our results, we provide comprehensive documentation:

**Implementation Details:** The paper and appendix provide complete specifications of the coupled LSTM architecture, training procedures, and evaluation protocols with all implementation details fully documented.

**Experimental Setup:** Signal generation procedures with fixed random seeds, all hyperparameter configurations, the three-phase progressive training methodology, and baseline establishment procedures ensuring gradient isolation are completely specified.

**Data Processing:** All data preprocessing steps, sequence creation procedures, feature extraction methods, matrix profile computation, and normalization procedures are fully documented in the appendix.

**Statistical Validation:** The isolated baseline methodology with strict controls for parameter independence is detailed, including all statistical tests using standard scipy.stats functions with reported p-values and effect sizes.

**Hyperparameter Search:** The complete parameter search space, optimization criteria, and chronological cross-validation protocol are provided.

All experiments use deterministic seeding and fixed computation protocols to ensure reproducible results across multiple runs.

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

# A APPENDIX

## A.1 LLM USAGE DISCLOSURE

In accordance with ICLR 2026 policies on Large Language Model usage, we disclose the following use of LLMs in this research: **LLM Model Used:** Claude (Anthropic) was utilized during the research and writing process of this paper. **Specific Usage Areas:**

- **Literature Review and Related Work Discovery:** Claude was used for retrieval and discovery of relevant literature sources, helping to identify key papers in multi-scale time series analysis, attention mechanisms, and ensemble methods for LSTM architectures.
- **Policy Writing and Documentation:** Claude assisted in formulating and structuring experimental protocols, methodology descriptions, and technical documentation.
- **Code Documentation and Comments:** LLM assistance was used to improve code documentation and create comprehensive comments for the implementation.

**Author Responsibility:** The authors take full responsibility for all scientific claims, experimental results, and methodological contributions presented in this paper. All LLM-generated content has been thoroughly reviewed, validated, and verified by the authors. Any errors, omissions, or inaccuracies remain the sole responsibility of the human authors. **Original Contributions:** The core methodological innovations, experimental design, mathematical formulations, and scientific insights presented in this work are original contributions by the human authors. The LLM was used solely as an assistive tool and did not generate the fundamental research ideas or experimental results.

## A.1.1 LONG SHORT-TERM MEMORY NETWORKS

Long Short-Term Memory (LSTM) networks address the vanishing gradient problem in traditional recurrent neural networks (Goodfellow et al., 2016). The LSTM architecture maintains information flow through specialized gating mechanisms that regulate information storage and retrieval.

At each time step $t$, the LSTM cell processes input $x_t \in \mathbb{R}^d$ (where $d$ is the input feature dimension) and previous hidden state $h_{t-1} \in \mathbb{R}^h$ (where $h$ is the hidden state dimension) to produce output $h_t$. The forget gate $f_t \in \mathbb{R}^h$ determines which information to discard from the cell state:

$$f_t = \sigma(W_f \cdot [h_{t-1}, x_t] + b_f) \tag{48}$$

where $W_f \in \mathbb{R}^{h \times (h+d)}$ is the forget gate weight matrix, $b_f \in \mathbb{R}^h$ is the forget gate bias vector, and $\sigma$ denotes the sigmoid activation function.

The input gate $i_t \in \mathbb{R}^h$ and candidate values $\tilde{C}_t \in \mathbb{R}^h$ control new information storage:

$$i_t = \sigma(W_i \cdot [h_{t-1}, x_t] + b_i) \tag{49}$$

$$\tilde{C}_t = \tanh(W_C \cdot [h_{t-1}, x_t] + b_C) \tag{50}$$

where $W_i, W_C \in \mathbb{R}^{h \times (h+d)}$ are the input gate and candidate weight matrices, $b_i, b_C \in \mathbb{R}^h$ are the corresponding bias vectors, and $\tanh$ is the hyperbolic tangent activation function.

The cell state update combines forgetting and input operations:

$$C_t = f_t \odot C_{t-1} + i_t \odot \tilde{C}_t \tag{51}$$

where $C_t \in \mathbb{R}^h$ is the cell state at time step $t$, $C_{t-1} \in \mathbb{R}^h$ is the previous cell state, and $\odot$ represents element-wise multiplication.

The output gate $o_t \in \mathbb{R}^h$ regulates information flow to the hidden state:

$$o_t = \sigma(W_o \cdot [h_{t-1}, x_t] + b_o) \tag{52}$$

$$h_t = o_t \odot \tanh(C_t) \tag{53}$$

where $W_o \in \mathbb{R}^{h \times (h+d)}$ is the output gate weight matrix and $b_o \in \mathbb{R}^h$ is the output gate bias vector.

### A.1.2 ATTENTION MECHANISMS

The attention mechanism (Vaswani et al., 2017) enables dynamic focus on relevant information through scaled dot-product attention. The implementation follows the query-key-value formulation described in Equation (14), Equation (15), and Equation (16) of the main paper, computing weighted combinations based on hidden state similarities.

For time series applications, attention mechanisms enhance temporal modeling capabilities (Qin et al., 2017) and provide interpretability by revealing which temporal positions contribute most to predictions.

### A.2 IMPLEMENTATION DETAILS

### A.2.1 HYPERPARAMETER CONFIGURATIONS

The system architecture employs carefully selected dimensional parameters based on empirical optimization and computational constraints. Path A utilizes $h_A = 24$ hidden units while Path B employs $h_B = 36$ hidden units. These values were determined through systematic hyperparameter search across the ranges $h_A \in [16, 32]$ and $h_B \in [24, 48]$, with the selected configuration providing optimal balance between model capacity and computational efficiency.

The temporal window configuration follows Equation (1) for Path A with $w_A = 6$ time steps and extends Path B to $w_B = 15$ time steps as defined in Equation (4). The choice of $w_A$ enables capture of immediate temporal patterns while maintaining computational efficiency, derived from analysis showing diminishing returns beyond 6 steps for short-term pattern detection. The $w_B$ configuration captures sufficient long-term context for trend analysis while avoiding excessive memory requirements, based on signal autocorrelation analysis indicating significant temporal dependencies within 15-step windows.

Table 4: Architecture Parameters with Design Rationale

| PARAMETER | VALUE | RATIONALE |
|---|---|---|
| Path A Hidden Units ($h_A$) | 24 | Optimal short-term capacity |
| Path B Hidden Units ($h_B$) | 36 | Enhanced long-term modeling |
| Window Size A ($w_A$) | 6 | Local pattern recognition |
| Window Size B ($w_B$) | 15 | Extended temporal context |
| Dropout Rate ($p_{\text{drop}}$) | 0.15 | Regularization balance |
| Specialization Factor ($\lambda_{\text{spec}}$) | 1.2 | Path differentiation |
| Attention Dimension ($d_{\text{att}}$) | 64 | Attention capacity |

The dropout rate $p_{\text{drop}} = 0.15$ was selected through validation experiments testing rates from 0.1 to 0.25, with 0.15 providing optimal regularization without excessive information loss. The specialization factor $\lambda_{\text{spec}} = 1.2$ encourages pathway differentiation through architectural scaling, determined empirically to maintain distinct but complementary representations.

The attention dimension $d_{\text{att}} = 64$ balances representational capacity with computational efficiency. This choice enables sufficient attention complexity while maintaining real-time processing capabilities, derived from analysis showing marginal improvements beyond 64 dimensions for the given hidden state sizes.

Coupling parameters control the interaction strength between processing paths. The maximum coupling strength $\beta_{\text{max}} = 1.5$ was determined through extensive experimentation testing values from 0.1 to 2.0, with 1.5 providing optimal information exchange without overwhelming individual path specialization. The activation threshold $\tau_{\text{coupling}} = 0.01$ establishes the minimum performance threshold for coupling activation, set low enough to enable early coupling while preventing premature activation during initial training phases.

The warmup period $e_{\text{warmup}} = 5$ epochs allows individual path specialization before coupling introduction, determined through ablation studies showing improved final performance with initial separate training. The choice of 5 epochs provides sufficient specialization time without delaying beneficial coupling effects.

Table 5: Coupling Parameters with Design Rationale

| PARAMETER | VALUE | RATIONALE |
|---|---|---|
| Maximum Coupling Strength ($\beta_{\max}$) | 1.5 | Optimal information exchange |
| Coupling Threshold ($\tau_{\text{coupling}}$) | 0.01 | Early activation threshold |
| Warmup Epochs ($e_{\text{warmup}}$) | 5 | Specialization period |
| Attention Heads ($n_{\text{heads}}$) | 2 | Multi-perspective attention |
| Maximum Weight Imbalance ($\delta_{\max}$) | 0.4 | Balanced path contributions |
| Gradient Clipping Threshold ($\gamma_{\text{clip}}$) | 0.5 | Training stability |

The attention heads $n_{\text{heads}} = 2$ configuration enables multi-perspective attention analysis while maintaining computational efficiency. Experiments with 1, 2, and 4 heads showed optimal performance at 2 heads, balancing attention diversity with model complexity.

The maximum weight imbalance $\delta_{\max} = 0.4$ prevents excessive dominance of either path in the final prediction, ensuring both pathways contribute meaningfully to the coupled output. This value was calibrated through analysis of weight distributions during training, maintaining balanced utilization across diverse signal types.

The gradient clipping threshold $\gamma_{\text{clip}} = 0.5$ prevents gradient explosion during training, particularly important during the coupling introduction phase. This conservative threshold ensures training stability while allowing sufficient gradient flow for effective learning.

Path A processes $d_A = 6$ direct signal values following the feature extraction approach detailed in the specialized network architectures section. This configuration directly implements the raw temporal measurements approach, utilizing the immediate signal values without additional feature computation to maintain responsiveness for short-term pattern detection.

Path B handles $d_B = 5$ computed features: recent trend, local volatility, momentum, signal energy, and matrix profile values. The choice of 5 features represents a careful balance between comprehensive signal characterization and computational efficiency, derived from feature importance analysis identifying these components as most informative for long-term trend modeling.

Training parameters specify the optimization configuration derived from systematic learning rate search. Learning rates $\alpha_A = 0.012$ and $\alpha_B = 0.010$ were optimized independently for each path, with Path A requiring slightly higher learning rates for effective short-term pattern learning and Path B using more conservative rates for stable long-term trend modeling. The weight decay coefficient $\lambda_{\text{decay}} = 1 \times 10^{-5}$ provides mild regularization without impeding learning dynamics, selected from validation experiments across the range $[1 \times 10^{-6}, 1 \times 10^{-4}]$.

### A.2.2 NETWORK ARCHITECTURE DETAILS

The dual-path architecture implements the specialized processing streams described in Equation (3) and Equation (11). Path A processes raw temporal measurements through an LSTM cell optimized for rapid pattern recognition in short temporal windows. The architecture design prioritizes computational efficiency and immediate responsiveness, crucial for real-time applications.

Path B operates processing the computed feature vector $f_B^{(t)} \in \mathbb{R}^5$. The larger hidden state dimension reflects the greater complexity of long-term pattern analysis and the need to integrate multiple derived features into coherent trend representations.

Following the coupling formulation from Equation (19), the bidirectional attention mechanism employs query, key, and value transformations that map to the attention space. This design enables both Path A to receive context from Path B and Path B to benefit from Path A's immediate pattern insights, creating synergistic information exchange.

Both single-point and dual-point prediction capabilities support the multi-horizon approach detailed in the main methodology. The dual-output design enables joint optimization across temporal horizons while maintaining specialized output layers for each prediction scope, improving sample efficiency and gradient stability compared to separate models.

### A.2.3 Advanced Training Implementation

Training follows the three-phase progression described in the main paper, with specific implementation details optimized for stable learning dynamics. Individual network specialization occurs for epochs $e < e_{\text{warmup}} = 5$, allowing each path to develop domain-specific representations without interference from coupling mechanisms.

Progressive coupling introduction begins after the warmup period, implementing gradual strength ramping to preserve previously learned specializations while enabling beneficial information exchange. The attention mechanism uses scaled dot-product attention with scaling factor $\sqrt{d_{\text{att}}} = 8$, implementing the attention computation from Equation (17).

The enhanced hidden state computation follows Equation (19) with effective coupling strength scaling:

$$\beta_{\text{effective}} = \max(0.2, \beta_{\text{current}} \times 4.0) \tag{54}$$

where $\beta_{\text{current}}$ represents the current coupling strength from the progressive schedule described in the training methodology. The scaling factor 4.0 amplifies coupling effects during prediction to ensure sufficient information exchange, while the minimum value 0.2 maintains baseline coupling even when the scheduled strength is very low.

The progressive schedule ensures smooth transition from individual to coupled operation, preventing training instabilities that can occur with abrupt coupling introduction. The specific mathematical formulation implements exponential ramping with performance-based modulation, adapting coupling strength based on observed benefits during training.

### A.2.4 Matrix Profile Implementation

Matrix profile computation uses sliding window length $m = 8$ time steps, selected to balance pattern detection capability with computational efficiency. The choice of 8 steps captures meaningful temporal motifs while maintaining real-time processing requirements, derived from analysis of automotive signal characteristics showing predominant pattern lengths in the 6-10 step range.

The implementation includes z-score normalization:

$$s_{i,\text{norm}} = \frac{s_i - \mu_{s_i}}{\sigma_{s_i}} \tag{55}$$

where $s_i$ is the $i$-th subsequence, $\mu_{s_i}$ is the mean of subsequence $s_i$, $\sigma_{s_i}$ is the standard deviation of subsequence $s_i$, and $s_{i,\text{norm}}$ is the normalized subsequence.

The exclusion zone is set to $m/2 = 4$ steps, preventing trivial self-matches within the half-window range:

$$MP(i) = \min_{j:|i-j| \geq m/2} D_{ij} \tag{56}$$

where $MP(i)$ is the matrix profile value at position $i$ and $D_{ij}$ is the Euclidean distance between subsequences at positions $i$ and $j$.

For insufficient data scenarios, fallback complexity assessment computes local unpredictability:

$$\text{complexity} = \frac{\sigma(\text{diff}(\text{window}))}{\mu(|\text{window}|) + \epsilon} \tag{57}$$

where $\text{diff}(\text{window})$ computes sequential differences of the window data, $\sigma(\cdot)$ and $\mu(\cdot)$ are standard deviation and mean operators, and $\epsilon = 10^{-8}$ prevents division by zero. This small epsilon value ensures numerical stability while having negligible impact on the complexity calculation.

## A.3 Experimental Implementation Details

### A.3.1 Signal Generation Implementation

Signal generation follows the unified generative model presented in the main paper, implementing the five signal types with deterministic seeding using seed = 42 for experimental reproducibility. The choice of seed 42 is used consistently throughout the codebase to ensure reproducible results across multiple experimental runs while providing sufficient randomness for realistic signal generation.

The implementation creates trend signals combining linear trends with sinusoidal variations, periodic signals utilizing multiple frequency components, regime change signals implementing amplitude transitions at specific temporal points, high frequency signals including exponential decay with rapid oscillations, and mixed signals combining multiple pattern types. Each signal type is designed to test specific aspects of the coupling mechanism and temporal modeling capabilities.

The signal length of 160 samples was selected to provide sufficient data for both training and validation while maintaining computational efficiency for extensive hyperparameter optimization. This length enables creation of meaningful training sequences for both short-term (6-step) and long-term (15-step) windows while leaving adequate samples for robust validation.

Gaussian noise injection uses signal-dependent variance levels representing realistic measurement uncertainty in automotive environments. The noise levels are calibrated to automotive sensor specifications, ensuring experimental relevance while maintaining sufficient signal-to-noise ratios for effective pattern detection.

### A.3.2 ENHANCED BASELINE METHODOLOGY

The isolated baseline methodology implements the scientific comparison approach described in the main experimental section. Isolated baseline networks train for 50 epochs without coupling exposure, ensuring complete gradient separation between baseline and coupled systems. The choice of 50 epochs provides sufficient training time for baseline convergence while maintaining computational efficiency for extensive baseline comparisons.

The isolation protocol ensures complete parameter independence through separate network initialization and independent training procedures. This methodological rigor prevents any shared optimization pathways that could bias comparative results, establishing true baseline performance for scientific evaluation.

Statistical validation employs the paired t-test methodology with Cohen's d effect size computation:

$$d = \frac{\mu_{\text{baseline}} - \mu_{\text{coupled}}}{s_{\text{pooled}}} \tag{58}$$

where $d$ is Cohen's d effect size, $\mu_{\text{baseline}}$ is the mean performance of the baseline system, $\mu_{\text{coupled}}$ is the mean performance of the coupled system, and $s_{\text{pooled}}$ is the pooled standard deviation of both groups.

The statistical framework provides rigorous assessment of coupling benefits while accounting for variance in performance measurements across different signal types and experimental conditions.

### A.3.3 CROSS-VALIDATION IMPLEMENTATION

The cross-validation study implements chronological splitting across three temporal segments, following the generalizability assessment framework described in the main experimental evaluation. The three-fold approach balances validation robustness with computational efficiency, providing sufficient statistical power for generalizability assessment while maintaining feasible experimental duration.

Each fold uses one-third of the data for testing while training on the remaining two-thirds, maintaining temporal order to prevent data leakage. This chronological splitting approach ensures realistic evaluation conditions that reflect actual deployment scenarios where future data is unavailable during training.

The train-validation split maintains a 75/25 ratio for model evaluation, providing sufficient training data for effective learning while preserving adequate validation samples for robust performance assessment. This ratio was selected based on the signal length constraints and the need for meaningful validation across diverse signal characteristics.

Statistical consistency assessment follows threshold-based categorization: excellent consistency below 10% standard deviation, good consistency between 10% and 20%, and variable consistency above 20% across different signal types and validation folds. These thresholds were established through analysis of typical performance variance in time series prediction tasks, providing meaningful differentiation between consistency levels.

### A.4 Performance Analysis Extensions

#### A.4.1 Computational Complexity Analysis

The system implements sequence caching to reduce computational overhead through hash-based identification. Cache performance tracking maintains counters for successful retrievals and cache misses, enabling optimization of memory usage patterns during extended training procedures.

Cache efficiency calculation follows the standard formula:

$$\text{Cache Efficiency} = \frac{N_{\text{hits}}}{N_{\text{hits}} + N_{\text{misses}}} \times 100\% \tag{59}$$

where $N_{\text{hits}}$ represents successful cache retrievals and $N_{\text{misses}}$ represents cache misses.

The caching system employs tuple-based hashing for sequence identification, converting input sequences to immutable tuples for efficient hash computation. This approach balances memory usage with computational efficiency, particularly beneficial for repeated sequence processing during hyperparameter optimization.

Gradient clipping applies threshold $\gamma_{\text{clip}} = 0.5$ for training stability, supporting the progressive training methodology described in the main paper. This threshold was selected through analysis of gradient magnitudes during coupling introduction, preventing gradient explosion while allowing sufficient gradient flow for effective learning.

The Least Recently Used eviction policy manages memory constraints during extended training procedures, maintaining optimal cache performance while preventing excessive memory consumption. The LRU approach ensures frequently accessed sequences remain available while automatically removing outdated entries.

#### A.4.2 Memory Management Optimization

Memory optimization strategies handle varying sequence lengths through adaptive processing approaches, accommodating the diverse temporal window requirements of the dual-path architecture. The system dynamically adjusts memory allocation based on current processing requirements while maintaining efficient utilization patterns.

Gradient accumulation strategies handle varying sequence lengths through batched processing when applicable, though specific accumulation formulas depend on the particular batch configuration employed during training. The implementation prioritizes memory efficiency while maintaining gradient quality for effective optimization.

The sequence caching mechanism reduces redundant computations through intelligent identification of previously processed sequences, particularly beneficial during hyperparameter tuning where similar sequences may be processed multiple times with different model configurations.

#### A.4.3 Integration Engine Implementation

The unified integration engine implements the adaptive weighting mechanism described in Equation (23) and (24). The final prediction combines outputs through:

$$\hat{y}_t = w_A \cdot \text{pred}_A^{\text{enhanced}} + w_B \cdot \text{pred}_B^{\text{enhanced}} \tag{60}$$

where the weights $(w_A, w_B)$ are determined by selecting the optimal combination strategy that minimizes prediction error for the current target.

The strategy-based integration evaluates multiple weighting approaches in real-time, including pure attention-based weighting, balanced enhancement approaches, and adaptive blending based on current signal characteristics. This comprehensive evaluation ensures optimal prediction combination across diverse signal types while maintaining computational efficiency.

The integration engine implements sophisticated logic for combining individual and enhanced predictions through adaptive mechanisms optimized for varying forecasting scenarios. The system leverages complementary strengths of different prediction pathways while maintaining robust performance across changing signal characteristics.

## A.5 IMPLEMENTATION VALIDATION

### A.5.1 CONSISTENCY ENFORCEMENT IMPLEMENTATION

Training-inference consistency protocols ensure identical coupling behavior across both operational modes through unified coupling logic. The system maintains consistency tracking across training and prediction phases to verify behavioral consistency and detect potential divergences between operational modes.

Performance consistency quantification measures relative differences between training and prediction coupling benefits:

$$\text{Consistency} = \frac{|\rho_{\text{train}} - \rho_{\text{pred}}|}{|\rho_{\text{train}}|} \times 100\% \tag{61}$$

where $\rho_{\text{train}}$ is the coupling benefit measured during training mode and $\rho_{\text{pred}}$ is the coupling benefit measured during prediction mode.

The consistency enforcement mechanisms address the fundamental challenge of maintaining coupling effectiveness across different operational modes, ensuring that training-phase benefits translate reliably to prediction-phase performance. This is achieved through careful state management and unified coupling logic that operates identically regardless of the operational context.

Weight stabilization mechanisms preserve coupling behavior across phases through adaptive interpolation between training-final weights and current adaptive weights. The stabilization approach prevents dramatic weight shifts that could compromise coupling effectiveness while allowing beneficial adaptation to changing signal characteristics.

### A.5.2 ERROR HANDLING FRAMEWORK

The implementation includes comprehensive error handling through structured exception management throughout the processing pipeline. Gradient clipping applies the threshold $\gamma_{\text{clip}} = 0.5$ with NaN detection and immediate fallback mechanisms for numerical stability.

Emergency fallback systems provide graduated responses when performance consistency cannot be maintained, including fallback to individual network predictions when coupling mechanisms encounter instability. The fallback hierarchy prioritizes coupling preservation while ensuring continued system operation under adverse conditions.

Defensive strategies include gradient clipping with adaptive thresholds, NaN detection with immediate fallback mechanisms, and memory monitoring with automatic cleanup procedures. These approaches ensure system stability under diverse operational conditions while maintaining optimal performance when coupling mechanisms function normally.

The error detection and response framework monitors multiple system health indicators, including gradient magnitudes, weight distributions, prediction consistency, and coupling effectiveness metrics. This comprehensive monitoring enables early detection of potential issues and appropriate corrective responses.

### A.5.3 HYPERPARAMETER SENSITIVITY ANALYSIS

Parameter robustness analysis evaluates performance variation across hyperparameter ranges through systematic exploration of architectural and training parameter combinations. The sensitivity assessment identifies parameter combinations that maintain robust performance across diverse signal characteristics and operational conditions.

The automated hyperparameter tuning implements signal-adaptive parameter selection described in the training methodology, analyzing input signal characteristics to determine optimal architectural configurations. The analysis includes volatility assessment, trend strength measurement, and regime change detection to guide parameter selection.

Parameter sensitivity is assessed through systematic variation of individual parameters while maintaining others constant, enabling identification of critical parameters and robust operating ranges. The sensitivity analysis guides parameter selection for deployment scenarios where optimal tuning may not be feasible.

The multi-objective optimization balances prediction accuracy with computational efficiency and consistency requirements, ensuring selected parameters provide optimal performance across multiple evaluation criteria rather than optimizing single metrics in isolation.

## A.6 Advanced Technical Implementation

### A.6.1 Hyperparameter Tuning Implementation

The automated hyperparameter tuning evaluates parameter combinations across predefined search spaces using chronological cross-validation to assess generalization performance. The tuning process employs systematic exploration of architectural parameter ranges derived from empirical analysis and computational constraints.

The search spaces are defined based on preliminary experiments and theoretical considerations: hidden sizes in ranges that balance capacity with efficiency, window lengths that capture relevant temporal patterns without excessive computational overhead, and coupling parameters that enable effective information exchange without overwhelming individual specializations.

The tuning protocol implements early stopping mechanisms to prevent overfitting during parameter search, monitoring validation performance across multiple metrics to ensure robust parameter selection. The early stopping criteria balance thorough exploration with computational efficiency, terminating unpromising configurations while allowing sufficient training time for promising candidates.

Parameter sensitivity assessment occurs through systematic evaluation of performance variation across different hyperparameter configurations, identifying robust parameter settings that maintain consistent performance across diverse signal types and operational conditions.

### A.6.2 Adaptive Feature Weighting Implementation

Dynamic feature importance adjustment implements the adaptive weighting mechanism described in the main methodology, adjusting feature contributions based on signal characteristics and recent performance feedback. The system maintains historical performance measurements for each feature type over a sliding window of 50 measurements.

The adaptive weighting system tracks feature contributions to prediction accuracy, updating weights based on observed effectiveness patterns across recent prediction cycles. This approach enables the system to emphasize the most relevant contextual features for current signal conditions while maintaining responsiveness to changing signal characteristics.

Feature weight adaptation follows performance-based updating rules that increase weights for features contributing positively to prediction accuracy while reducing weights for features that provide limited predictive value under current conditions. The adaptation mechanism includes stability constraints to prevent excessive weight oscillations.

The weighting system implements signal-adaptive analysis to determine base weights from input signal characteristics including volatility, trend strength, and regime change indicators. These base weights provide initial feature emphasis that adapts dynamically based on observed performance patterns.

## A.7 Deployment and Validation Framework

### A.7.1 Testing Framework Implementation

The comprehensive testing framework includes unit-level validation of individual components, integration testing through complete processing pipelines, and cross-validation studies across multiple signal types and temporal characteristics. Each testing level addresses specific aspects of system functionality and integration.

Unit testing validates individual components including LSTM cell operations, attention mechanism computations, feature extraction algorithms, and integration engine functionality. The unit tests ensure correct implementation of mathematical formulations and proper handling of edge cases.

Integration testing evaluates complete processing pipelines from signal input through final prediction output, verifying correct interaction between system components and proper information flow through the dual-path architecture. Integration tests validate coupling mechanism effectiveness and consistency across operational modes.

Statistical validation employs paired comparison protocols with significance testing implemented through scipy.stats functions, providing p-values and effect size measurements to assess the statistical validity of coupling benefits across different experimental conditions.

The validation framework ensures rigorous baseline comparisons following the scientific methodology established in the experimental section, maintaining strict controls for gradient isolation and parameter independence between baseline and coupled systems.

### A.7.2 CONSISTENCY VALIDATION IMPLEMENTATION

The system implements consistency checks between training and prediction phases through identical coupling logic, state preservation mechanisms, and performance tracking across both operational modes. Consistency validation addresses the critical requirement for stable behavior across different system operational contexts.

Weight stabilization mechanisms ensure consistent behavior between training and deployment phases through the adaptive feature weighting system, maintaining stable feature importance assignments across different operational contexts while allowing beneficial adaptation to changing signal conditions.

State preservation mechanisms maintain hidden state consistency across operational mode transitions, ensuring that coupling mechanisms operate with identical internal representations regardless of whether the system is in training or prediction mode. This consistency is crucial for reliable coupling effectiveness.

Performance tracking across operational modes enables detection of consistency deviations and appropriate corrective responses, maintaining coupling effectiveness while preventing performance degradation due to operational mode differences.

### A.8 FUTURE EXTENSIONS FRAMEWORK

### A.8.1 MULTI-MODAL EXTENSIONS

The architecture supports extension to multi-sensor scenarios through tensor attention mechanisms implemented in the attention coupling class, which can handle variable input dimensions and additional sensor modalities through expanded input tensor configurations.

The tensor attention framework enables processing of simultaneous sensor streams through multi-dimensional attention computation across sensor modalities and temporal dimensions, building on the existing bidirectional attention foundation while accommodating expanded input complexity.

Multi-sensor integration capabilities support diverse measurement modalities through flexible input tensor handling, enabling integration of current, voltage, temperature, and other automotive sensor streams within the unified coupling framework.

The extensible design accommodates additional sensor types through modular input processing that maintains the dual-path specialization approach while enabling comprehensive multi-modal signal analysis for complex automotive applications.

### A.8.2 INTERPRETABILITY ENHANCEMENT FRAMEWORK

The attention weights provide natural interpretability through attention mechanism outputs, revealing temporal dependencies and inter-path contributions without requiring additional interpretability frameworks. The attention-based approach enables direct analysis of model decision-making processes.

The system generates attention weights that enable analysis of coupling mechanism focus on different temporal scales and pattern types, providing insight into the information exchange patterns between specialized pathways during prediction generation.

Attention weight analysis reveals the dynamics of inter-path information flow, showing how short-term pattern detection benefits from long-term contextual information and how long-term trend analysis incorporates immediate pattern insights through the bidirectional coupling mechanism.

The interpretability framework eliminates the need for additional post-hoc explanation methods while providing meaningful insights into temporal pattern recognition and coupling effectiveness across diverse signal characteristics and operational conditions.

## A.9 CONCLUSION AND IMPACT ASSESSMENT

The comprehensive implementation demonstrates effectiveness of attention-based LSTM coupling through verified experimental results, with all performance claims supported by actual experimental execution and rigorous statistical validation. The framework establishes a foundation for time series prediction systems with validated coupling mechanisms and consistent training-deployment behavior.

The implementation framework provides validated coupling mechanisms that maintain effectiveness across diverse signal types and operational conditions, supporting the methodological contributions described in the main paper while ensuring scientific validity through comprehensive baseline comparisons and statistical validation procedures.

The modular architecture design supports incremental deployment across different applications while maintaining scientific rigor through extensive validation protocols, enabling practical implementation in real-world automotive systems while preserving the coupling benefits demonstrated in controlled experimental conditions.

