# OpenReview forum: "Adaptive Multi-Scale Attention-Based LSTM Coupling for Early Detection"
_ICLR.cc/2026/Conference — Submitted to ICLR 2026_

### Official Review · Reviewer_x4J2 · 2025-10-26

**Soundness:** 2
**Presentation:** 2
**Contribution:** 1
**Rating:** 0
**Confidence:** 4

**Summary:**

The paper proposes an adaptive attention-coupled LSTM architecture. This architecture employs two separate LSTMs to process data with different window lengths, thereby capturing information across distinct temporal domains. It integrates information from the two pathways via cross-attention with adaptive weights and generates predictions for multiple time horizons. Specifically designed for real-time time-series forecasting and scenario detection in complex automotive electrical/electronic (E/E) systems.

**Strengths:**

1.Clear Methodological Logic. The core idea of the adaptive attention-coupled LSTM is straightforward and well-structured. The technical route from problem definition to solution implementation is logical and easy to follow.
2.Tailored Weight Parameter Schemes for Different Scenarios. The paper designs distinct weight parameter strategies for various application scenarios. These scenario-specific weight designs enhance the method’s adaptability to diverse temporal patterns in automotive E/E system data.

**Weaknesses:**

1.The proposed method has not been validated through experiments on other types of data.
2.Lack of Justification for the Addressed Problem. The paper fails to sufficiently demonstrate the necessity and urgency of the problem it claims to solve—like the limitations of existing methods in real-time time-series forecasting for automotive E/E systems.
3.Limited Innovation and Oversimplified Methodology. The adaptive attention-coupled LSTM primarily combines existing techniques (dual-pathway LSTMs + cross-attention) without introducing fundamental innovations.
4.No Demonstration of Runtime Efficiency for Real-Time Scenarios. While the paper claims the method is designed for "real-time scenarios," it provides no quantitative data on runtime efficiency.
5.Insufficient Experiments and Lack of Baseline Comparisons. The experimental scope is narrow, and there is a lack of comparative data with mainstream baseline methods. For example:  The paper does not include benchmarks widely used in automotive time-series forecasting; It fails to validate performance on publicly available automotive E/E dataset, relying instead on potentially custom or synthetic data.
6.Low Paper Completion. No Clear Architectural Schematic. The paper lacks explicit architectural schematic diagrams and related works.
7.Narrow Application Scope. The method is exclusively designed for automotive E/E systems and has not been validated on other time-series scenarios. This limits the method’s generalizability and academic impact.
8.Lack of Ablation Studies and Parameter Sensitivity Analysis. There are no ablation experiments to validate the effectiveness of key components (e.g., adaptive weights, cross-attention) or analyze the impact of hyperparameters (e.g., time window length, number of hidden units). It is impossible to determine whether the improved performance stems from the proposed design or merely increased model parameters.

**Questions:**

1.More experiments are needed.
2. See weakness.

---

### Official Review · Reviewer_thjK · 2025-10-28

**Soundness:** 3
**Presentation:** 2
**Contribution:** 3
**Rating:** 2
**Confidence:** 3

**Summary:**

This paper proposes an Adaptive Multi-Scale Attention network for time series forecasting, specifically motivated by early anomaly detection and scenario prediction in automotive systems.
The main idea is to separate short-term and long-term dependencies into 2 specialized LSTM branches (“dual-path”), which exchange information through bidirectional scaled dot-product attention.
A 3-stage training schedule (specialization, coupling ramp-up, joint refinement) is introduced, along with an adaptive feature-weighting mechanism to emphasize dynamically relevant inputs.
Experiments on a synthetic dataset show a significant reduction in Mean Squared Error compared to isolated LSTM baselines.
While the architecture is conceptually sound and intuitively appealing, the novelty is not clearly demonstrated with respect to prior multi-scale or hierarchical LSTM approaches, and the evaluation is restricted to synthetic data. The paper would benefit from clearer theoretical justification and improved reproducibility.

**Strengths:**

- The idea of decoupling short and long-term temporal patterns and coupling them via attention is intuitive and could have practical advantages.
- The 3-phase schedule is well thought out and potentially generalizable.
- The reported improvements in MSE demonstrate the model’s potential effectiveness in controlled scenarios.
- The paper includes implementation information, which, if integrated into the main text, could help reproducibility and lead to a better understanding of the approach.

**Weaknesses:**

- The paper could benefit from better motivation.
- The authors do not clearly differentiate their proposal from prior work, particularly in the Related Work section.
- All results are based on synthetic data with Gaussian noise, and there is no testing on real or public benchmarks, which limits the applicability and impact of the findings.
- No code or data repository has been provided.
- There are contradictions within the paper, especially regarding the notation and ranges for hyperparameters.
- The description of the methodology is vague and contains excessive repetition of certain concepts (e.g., dual path or short vs. long). Some explanations are found only in the appendix. A more comprehensive description of the proposal should be included in the main body of the paper.

**Questions:**

1) Please clarify the discrepancy regarding β_max: the main text limits it to [0.1, 0.8], while Appendix A.2.1 lists 1.5 as optimal and 2.0 as the maximum limit.
2) The main text uses “d” for the dropout rate, whereas the appendix uses p_drop. Please confirm which notation is correct.
3) Regarding adaptive windowing: Section 3.3 mentions averaging over 10 steps, while Appendix A.6.2 refers to 50 steps. What is the actual configuration used in the experiments?
4) How exactly does LSTM A “dynamically adapt”? The methodology is not clearly explained.
5) What optimization strategy was utilized for hyperparameter optimization?
6) Considering that the dataset is synthetic, can this approach be tested on a publicly available time series dataset?
7) Many important methodological choices (e.g., preprocessing steps, window sizes, data generation parameters, hyperparameter ranges) are only located in the appendices. Could the authors summarize or move this information to the main text to clarify the overall pipeline and justify these design choices?

---

### Official Review · Reviewer_dXjN · 2025-10-30

**Soundness:** 3
**Presentation:** 3
**Contribution:** 3
**Rating:** 6
**Confidence:** 2

**Summary:**

This paper addresses the core challenges of real-time scene recognition in complex automotive electronic and electrical systems and proposes a breakthrough solution - the adaptive attention Coupled LSTM architecture. This study has for the first time systematically constructed a dual-path time series modeling paradigm that combines specialization and collaboration, fundamentally resolving the inherent contradiction of a single time series model in capturing long and short time dependencies. The core theoretical innovation lies in the introduction of a bidirectional and adaptive attention coupling mechanism, which serves as an "intelligent information hub" connecting the two specialized paths. To achieve the optimal collaborative performance, the author designed a progressive multi-stage training protocol. Through independent expertise cultivation, progressive coupling introduction, and global joint optimization, it ensures that the model is both highly specialized and collaborative. There exist some issues to be addressed as follows.

**Strengths:**

1.The paper addresses the core challenges of real-time scene recognition in complex automotive electronic/electrical (E/E) systems, proposing a novel and breakthrough solution—the adaptive attention-coupled LSTM architecture.

2.It represents the first systematic construction of a dual-path time series modeling paradigm that effectively combines specialization and collaboration, fundamentally resolving the inherent contradiction in single models for capturing both long- and short-term dependencies.

3.The core theoretical innovation lies in the introduction of a bidirectional and adaptive attention coupling mechanism, which acts as an "intelligent information hub" to dynamically connect the two specialized paths, enhancing information exchange.

4.The authors designed a progressive multi-stage training protocol (including independent expertise cultivation, gradual coupling introduction, and global joint optimization) to ensure optimal collaborative performance, balancing specialization and integration.

**Weaknesses:**

1.There is a lack of experimental data or justification for the design choices regarding the number of hidden units in LSTM A (24-32) and LSTM B (32-48). The paper does not provide empirical evidence or ablation studies to validate why these specific ranges were selected.

2.The paper fails to compare the proposed model with recent state-of-the-art time series prediction architectures, such as Transformer-based models (e.g., Informer, FEDformer), which have demonstrated advantages in capturing multi-scale dependencies. This omission limits the comprehensiveness of the evaluation.

3.The computational complexity and memory usage of the dual LSTM paths, combined with the attention mechanism, are not thoroughly analyzed. There is no deployment feasibility study or lightweight experiments for real-time inference on in-vehicle embedded devices, raising concerns about practical applicability.

4.The experiments are entirely based on synthetic data and lack validation on real-world automotive E/E system data. This reduces the reliability and generalizability of the results for actual automotive applications.

**Questions:**

1. Is there any experimental data to verify why the number of 24-32 and 32-48 hidden units, respectively, used by LSTM A and LSTM B was designed in this way?

2. Why not compare it with the time series prediction architectures that have performed well in recent years, such as Transformer-based models (such as Informer, FEDformer)? These models also have their advantages in capturing multi-scale dependencies.

3. How much does the computational complexity and memory usage of dual LSTM paths combined with the attention mechanism increase compared to a single LSTM baseline? Is there a deployment feasibility analysis or a lightweight experiment for real-time inference of in-vehicle embedded devices?

4. The experiments are entirely based on synthetic data and have not been verified on real automotive E/E system data.

---

### Official Review · Reviewer_emjW · 2025-10-31

**Soundness:** 2
**Presentation:** 2
**Contribution:** 2
**Rating:** 2
**Confidence:** 4

**Summary:**

The paper proposes a deep learning based time-series prediction model comprising two data encoding towers, one based on short-term data (past 6 time-steps) and another based on long-term data (past 15 time-steps). The final prediction is based on an attention coupling of predictions from the short-term and long-term towers to obtain the `enhanced` final prediction. The authors evaluate the proposed method on synthetic data that they have generated.

**Strengths:**

1. The paper proposes a method for an important problem of scenario recognition and prediction in the automotive system context.

2. Overall, the de-coupled modeling approach to estimate the short-term and long-term patterns are intuitive and the attention based mechanism to couple representations from the two towers is somewhat novel.

**Weaknesses:**

1. The paper is difficult to understand and lacks cohesion as many of the architecture choices seem somewhat arbitrary and scattered across sections 3 and 4. The paper (especially the methodology description) would benefit from a significant re-write. Several sections in the appendix can be eliminated. For example A.1.1 and A.1.2 are unnecessary as the machine learning audience is familiar with recurrent architectures like the LSTM and the attention mechanisms.


2. Overall, the paper lacks rigorous evaluation on real-world data and has only been evaluated on synthetic data. Further, the dataset generation procedure has not been described in detail.

3. There is no rigorous baseline comparison with other popular baselines e.g., state-space models like MAMBA.

**Questions:**

1. How are $w_{i,base}$ and $\gamma_i^{(t)}$ estimated? The text says $\gamma_i^{(t)}$ reflects the "average contribution" of each feature but it is unclear how this is derived / what "contribution" means.

2. How is $\beta^t$ estimated?

3. What are the various hyper-parameters that need to be tuned if the proposed method is to be adapted to a new dataset?

4. Why has no comparison been conducted with state of the art baselines (e.g., Mamba [1], standard autoregressive based and single-layer linear [2] models) which have shown effective performance in time-series forecasting scenarios?

5. Why is there no ablation study conducted to highlight the importance of various components (e.g., short-term, long-term and attention based components)? This is imperative to holistically understand how the various components contribute to the overall performance.

6. Are there any real-world datasets on the automobile real-time scenario recognition and prediction use-case that can be employed to test the effectiveness of the proposed method? Synthetic data is useful but cannot serve as a comprehensive evaluation of the performance of the proposed method.

# References

1. Wang, Zihan, Fanheng Kong, Shi Feng, Ming Wang, Xiaocui Yang, Han Zhao, Daling Wang, and Yifei Zhang. "Is mamba effective for time series forecasting?." Neurocomputing 619 (2025): 129178.

2. Zeng, Ailing, Muxi Chen, Lei Zhang, and Qiang Xu. "Are transformers effective for time series forecasting?." In Proceedings of the AAAI conference on artificial intelligence, vol. 37, no. 9, pp. 11121-11128. 2023.

---

### Meta-Review · Area_Chair_jLk6 · 2026-01-07

**Summary:**

The paper proposes an "Adaptive Attention-Coupled LSTM" architecture designed for real-time scenario recognition and forecasting in automotive E/E systems. The core methodology involves a dual-path LSTM framework: a "Pattern Path" (short-term window) and a "Context Path" (long-term window), connected via a bidirectional attention gating mechanism. A three-phase training protocol (independent specialization $\to$ progressive coupling $\to$ joint optimization) is introduced to balance the learning objectives. The authors evaluate the method on synthetically generated datasets representing various signal types (trend, periodic, regime change, etc.).

Strengths

- Intuitive Architecture: The decoupling of short-term motif detection and long-term trend modeling into specialized paths is a logical approach for multi-scale time series analysis.

- Training Schedule: The proposed three-phase progressive training protocol is well-motivated and potentially useful for training complex, multi-component architectures.

- Performance on Controlled Data: The method demonstrates significant MSE reductions compared to isolated LSTM baselines on the generated synthetic datasets.

Weaknesses
- Synthetic Evaluation: A critical flaw identified by all reviewers (emjW, dXjN, thjK, x4J2) is the exclusive reliance on synthetic data. For a paper targeting real-world automotive applications, the lack of validation on real-world datasets (automotive or public benchmarks) severely limits the impact and reliability of the findings.
- Insufficient Baselines: The paper compares primarily against isolated/ablated versions of itself. It lacks comparisons with modern state-of-the-art time series architectures such as Transformers (Informer, Autoformer) or State Space Models (Mamba), which are standard baselines for this domain.
- Lack of Ablation: There are no comprehensive ablation studies to quantify the specific contributions of key components (e.g., the attention mechanism vs. the dual-path structure vs. the adaptive weighting). It is unclear if the performance gains stem from the novel architecture or simply increased parameter count.
- Clarity and Inconsistencies: Reviewers noted significant discrepancies between the main text and appendix (e.g., hyperparameter ranges for $\beta_{max}$ and dropout notation). Critical methodological details are buried in the appendix or described vaguely in the main text.
- Operational Feasibility: Despite claiming applicability for "real-time" automotive systems, there is no analysis of computational complexity, latency, or memory usage on embedded hardware.JustificationThe submission cannot be accepted in its current form due to fundamental deficiencies in experimental validation. While the proposed dual-path architecture is conceptually interesting, the exclusive reliance on synthetic data is insufficient for an ICLR contribution, particularly one claiming practical relevance to automotive engineering.
- The absence of comparisons against modern SOTA baselines (Transformers, Mamba) and the lack of ablation studies make it impossible to assess the true novelty and effectiveness of the method. The paper requires a rigorous evaluation on real-world benchmarks, a clearer methodological description, and a direct comparison with established state-of-the-art models.

**Reviewer Concerns:**

Please see above

**Reviewer Scores:**

There was no rebuttal submitted to change reviewers' scores

---

### Decision · Program_Chairs · 2026-01-26

Reject